# Neural networks grown and self-organized by noise

**Guruprasad Raghavan**
Department of Bioengineering
Caltech
Pasadena, CA 91125
graghava@caltech.edu

**Matt Thomson**
Biology and Biological Engineering
Caltech
Pasadena, CA 91125
mthomson@caltech.edu

## Abstract

Living neural networks emerge through a process of growth and self-organization that begins with a single cell and results in a brain, an organized and functional computational device. Artificial neural networks, however, rely on human-designed, hand-programmed architectures for their remarkable performance. Can we develop artificial computational devices that can grow and self-organize without human intervention? In this paper, we propose a biologically inspired developmental algorithm that can 'grow' a functional, layered neural network from a single initial cell. The algorithm organizes inter-layer connections to construct retinotopic pooling layers. Our approach is inspired by the mechanisms employed by the early visual system to wire the retina to the lateral geniculate nucleus (LGN), days before animals open their eyes. The key ingredients for robust self-organization are an emergent spontaneous spatiotemporal activity wave in the first layer and a local learning rule in the second layer that 'learns' the underlying activity pattern in the first layer. The algorithm is adaptable to a wide-range of input-layer geometries, robust to malfunctioning units in the first layer, and so can be used to successfully grow and self-organize pooling architectures of different pool-sizes and shapes. The algorithm provides a primitive procedure for constructing layered neural networks through growth and self-organization. We also demonstrate that networks grown from a single unit perform as well as hand-crafted networks on MNIST. Broadly, our work shows that biologically inspired developmental algorithms can be applied to autonomously grow functional 'brains' in-silico.

## 1 Introduction

Living neural networks in the brain perform an array of computational and information processing tasks including sensory input processing [1, 2], storing and retrieving memory [3, 4], decision making [5, 6], and more globally, generate the general phenomena of "intelligence". In addition to their information processing feats, brains are unique because they are computational devices that actually self-organize their intelligence. In fact brains ultimately grow from single cells during development. Engineering has yet to construct artificial computational systems that can self-organize their intelligence. In this paper, inspired by neural development, we ask how artificial computational devices might build themselves without human intervention.

Deep neural networks are one of the most powerful paradigms in Artificial Intelligence. Deep neural networks have demonstrated human-like performance in tasks ranging from image and speech recognition to game-playing [7, 8, 9]. Although the layered architecture plays an important role in the success [10] of deep neural networks, the widely accepted state of art is to use a hand-programmed network architecture [11] or to tune multiple architectural parameters, both requiring significant engineering investment. Convolutional neural networks, a specific class of DNNs, employ a hand

programmed architecture that mimics the pooling topology of neural networks in the human visual system.

Here, we develop strategies for *growing a neural network* autonomously from a single computational "cell" followed by *self-organization* of its architecture by implementing a wiring algorithm inspired by the development of the mammalian visual system. The visual circuitry, specifically the wiring of the retina to the lateral geniculate nucleus (LGN) is stereotypic across organisms, as the architecture always enforces pooling (retinal ganglion cells (RGC's) pool their inputs to LGN cells) and retinotopy. The pooling architecture (figure-1a) is robustly established early in development through the emergence of spontaneous activity waves (figure-1b) that tile the light insensitive retina [12]. As the synaptic connectivity between the different layers in the visual system get tuned in an activity-dependent manner, the emergent activity waves serve as a signal to alter inter-layer connectivity much before the onset of vision.

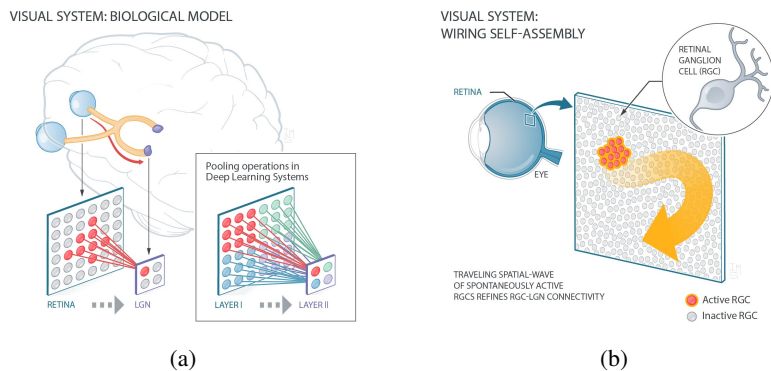

(a)             (b)

Figure 1: **Wiring of the visual circuitry** (a) Spatial pooling observed in wiring from the retina to LGN and in CNN's. (b) Synchronous Spontaneous bursts (retinal waves) in the light-insensitive retina serve as a signal for wiring retina to the brain.

The main contribution of this paper is that we propose a developmental algorithm inspired by visual system development to *grow and self-organize a retinotopic pooling architecture*, similar to modern convolutional neural networks (CNNs). Once a pooling architecture emerges, any non-linear function can be implemented by units in the second layer to morph it into functioning as a convolution or a max/average pooling. We show that our algorithm is adaptable to a wide-range of input-layer geometries, robust to malfunctioning units in the first layer and can grow pooling architectures of different shapes and sizes, making it capable of countering the key challenges accompanying growth. We also demonstrate that 'grown' networks are functionally similar to that of hand-programmed pooling networks, on conventional image classification tasks. As CNN's represent a model class of deep networks, we believe the developmental strategy can be broadly implemented for the self-organization of intelligent systems.

## 2 Related Work

Computational models for self-organizing neural networks dates back many years, with the first demonstration being Fukushima's neocognitron [13, 14], a hierarchical multi-layered neural network capable of visual pattern recognition through learning. Although weights connecting different layers were modified in an unsupervised fashion, the network architecture was hard-coded, inspired by Hubel and Wiesel's [15] description of simple and complex cells in the visual cortex. Fukushima's neocognitron inspired modern-day convolutional neural networks (CNN) [16]. Although CNN's performed well on image-based tasks, they had a fixed, hand-designed architecture whose weights were altered by back-propagation. The use of a fixed, hand-designed architecture for a neural network changed with the advent of neural architecture search [17], as neural architectures became malleable to tuning by neuro-evolution strategies [18, 19, 20], reinforcement learning [21] and multi-objective searches [22, 23]. Neuro-evolution strategies have been successful in training networks that perform significantly much better on CIFAR-10, CIFAR-100 and Image-Net datasets. As the objective function being maximized is the predictive performance on a single dataset, the evolved networks

may not generalize well to multiple datasets. On the contrary, biological neural networks in the brain grow architecture that can generalize very well to innumerable datasets. Neuroscientists have been very interested in how the architecture in the visual cortex emerges during brain development. Meister et al [12] suggested that spontaneous and spatially organized synchronized bursts prevalent in the developing retina guide the self-organization of cortical receptive fields. In this light, mathematical models of the retina and its emergent retinal waves were built [24], and analytical solutions were obtained regarding the self-organization of wiring between the retina and the LGN [25, 26, 27, 28, 29]. Computational models have been essential for understanding how self-organization functions in the brain, but haven't been generalized to growing complex architectures that can compute. One of the most successful attempts at growing a 3D model of neural tissue from simple precursor units was demonstrated by Zubler et. al [30] that defined a set of minimal rules that could result in the growth of morphologically diverse neurons. Although the networks were grown from single units, they weren't functional as they weren't equipped to perform any task. To bridge this gap, in this paper we attempt to grow and self-organize functional neural networks from a single precursor unit.

# 3  Bio-inspired developmental algorithm

In our procedure, the pooling architecture emerges through two processes, growth of a layered neural network followed by self-organization of its inter-layer connections to form defined 'pools' or receptive fields. As the protocol for growing a network is relatively straightforward, our emphasis in the next few sections is on the self-organization process, following which we will combine the growth of a layered neural network with its self-organization in the penultimate section of this paper.

We, first, abstract the natural development strategy as a mathematical model around a set of input sensor nodes in the first layer (similar to retinal ganglion cells) and processing units in the second layer (similar to cells in the LGN).

Self-organization comprises of two major elements: (1) A **spatiotemporal wave generator** in the first layer driven by noisy interactions between input-sensor nodes and (2) A **local learning rule** implemented by units in the second layer to learn the "underlying" pattern of activity generated in the first layer. The two elements are inspired by mechanisms deployed by the early visual system. The retina generates spontaneous activity waves that tile the light-insensitive retina; the activity waves serve as input signals to wire the retina to higher visual areas in the brain [31, 32].

## 3.1  Spontaneous spatiotemporal wave generator

The first layer of the network can serve as a noise-driven spatiotemporal wave generator when (1) its constituent sensor-nodes are modeled via an appropriate dynamical system and (2) when these nodes are connected in a suitable topology. In this paper, we model each sensor node using the classic Izikhevich neuron model [33] (dynamical system model), while the input layer topology is that of local-excitation and global-inhibition, a motif that is ubiquitous across various biological systems [34, 35]. A minimal dynamical systems model coupled with the local-excitation and global-inhibition motif has been analytically examined in the supplemental materials to demonstrate that these key ingredients are *sufficient* to serve as a spatiotemporal wave generator.



Figure 2: **Emergent spatiotemporal waves tile the first layer.** The red-nodes indicate active-nodes (firing), black nodes refer to silent nodes and the arrows denote the direction of time.

The **Izhikevich model** captures the activity of every sensor node ($v_i(t)$) through time, the noisy behavior of individual nodes (through $\eta_i(t)$) and accounts for interactions between nodes defined by a synaptic adjacency matrix ($S_{i,j}$). The Izhikevich model equations are elaborated in Box-1. The **input layer topology** (local excitation, global inhibition) is defined by the synaptic adjacency matrix ($S_{i,j}$). Every node in the first layer makes excitatory connections with nodes within a defined local excitation radius. $S_{i,j} = 5$, when distance between nodes $i$ and $j$ are within the defined excitation

radius of 2 units; $d_{ij} \leq 2$. Each node has decaying inhibitory connections with other nodes present above a defined global inhibition radius ($S_{i,j}$ = -2 exp(-$d_{ij}$/10), when distance between nodes $i$ and $j$ are above a defined inhibition radius of 4 units; $d_{ij} \geq 4$) (see supporting information).

On implementing a model of the resulting dynamical system, we observe the emergence of spontaneous spatiotemporal waves that tile the first layer for specific parameter regimes (see figure 2 and videos in supplemental materials).

---

**Dynamical model for input-sensor nodes in the lower layer (layer-I):**

$$\frac{\mathrm{d}v_i}{\mathrm{d}t} = 0.04v_i^2 + 5v_i + 140 - u_i + \sum_{j=1}^{N} S_{i,j}\mathcal{H}(v_j - 30) + \eta_i(t) \tag{1}$$

$$\frac{\mathrm{d}u_i}{\mathrm{d}t} = a_i(b_i v_i - u_i) \tag{2}$$

with the auxiliary after-spike reset:

$$v_i(t) > 30, \text{then} : \begin{cases} v_i(t + \Delta t) = c_i \\ u_i(t + \Delta t) = u_i(t) + d_i \end{cases}$$

where: (1) $v_i$ is the activity of sensor node $i$; (2) $u_i$ captures the recovery of sensor node $i$; (3) $S_{i,j}$ is the connection weight between sensor-nodes $i$ and $j$; (4) $N$ is the number of sensor-nodes in layer-I; (5) Parameters $a_i$ and $b_i$ are set to 0.02 and 0.2 respectively, while $c_i$ and $d_i$ are sampled from the distributions $\mathcal{U}(-65, -50)$ and $\mathcal{U}(2, 8)$ respectively. Once set for every node, they remain constant during the process of self-organization. The initial values for $v_i(0)$ and $u_i(0)$ are set to -65 and -13 respectively for all nodes. These values are taken from Izhikevich's neuron model [33]; (6) $\eta_i(t)$ models the noisy behavior of every node $i$ in the system, where $< \eta_i(t)\eta_j(t') > = \sigma^2 \delta_{i,j}\delta(t - t')$. Here, $\delta_{i,j}$, $\delta(t - t')$ are Kronecker-delta and Dirac-delta functions respectively, and $\sigma^2 = 9$; (7) $\mathcal{H}$ is the unit step function:

$$\mathcal{H}(v_i - 30) = \begin{cases} 1, & v_i \geq 30 \\ 0, & v_i < 30. \end{cases}$$

---

## 3.2 Local learning rule

Having constructed a spontaneous spatiotemporal wave generator in layer-I, we implement a local learning rule in layer-II that can learn the activity wave pattern in the first layer and modify its inter-layer connections to generate a pooling architecture. Many neuron inspired learning rules can learn a sparse code from a set of input examples [36]. Here, we model processing units as rectified linear units (ReLU) and implement a modified Hebbian rule for tuning the inter-layer weights to achieve the same. Individual ReLU units compete with one another in a winner take all fashion.

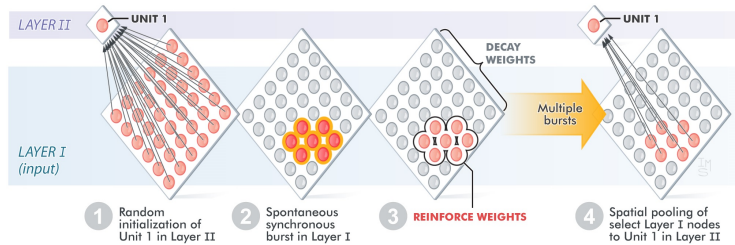

Figure 3: Learning rule

Initially, every processing unit in the second layer is connected to all input-sensor nodes in the first layer. As the emergent activity wave tiles the first layer, at most a single processing unit in the second layer is activated due to the winner-take-all competition. The weights connecting the activated unit in the second layer to the input-sensor nodes in the first layer are updated by the modified Hebbian rule

(Box-2). Weights connecting active input-sensor nodes and activated processing units are reinforced while weights connecting inactive input-sensor nodes and activated processing units decay (cells that fire together, wire together). Inter-layer weights are updated continuously throughout the self-organization process, ultimately resulting in the pooling architecture (See figure-3 and supplemental materials).

---

**Modifying inter-layer weights**

$$w_{i,j}(t+1) = \begin{cases} w_{i,j}(t) + \eta_{learn}\mathcal{H}(v_i(t) - 30)y_j(t+1) & y_j(t+1) > 0 \\ w_{i,j}(t) & \text{otherwise} \end{cases}$$

where: (1) $w_{i,j}(t)$ is the weight of connection between sensor-node $i$ and processing unit $j$ at time 't' (inter-layer connection); (2) $\eta_{learn}$ is the learning rate; (3) $\mathcal{H}(v_i(t) - 30)$ is the activity of sensor node $i$ at time 't'; and (4) $y_j(t)$ is the activation of processing unit $j$ at time 't'.

Once all the weights $w_{i,j}(t+1)$ have been evaluated for a processing unit $j$, they are mean-normalized to prevent a weight blow-up. Mean normalization ensures that the mean strength of weights for processing unit $j$ remains constant during the self-organization process.

---

Having coupled the spontaneous spatiotemporal wave generator and the local learning rule, we observe that an initially fully connected two-layer network (figure-4a) becomes a pooling architecture, wherein input-sensor nodes that are in close proximity to each other in the first layer have a very high probability of connecting to the same processing unit in the second layer (figure-4b & 4c). More than 95% of the sensor-nodes in layer-I connect to processing units in layer-II (higher layer) through well-defined pools, ensuring that spatial patches of nodes connected to units in layer-II tile the input layer (figure-4d). Tiling the input layer ensures that most sensor nodes have an established means of sending information to higher layers after the self-organization of the pooling layer.

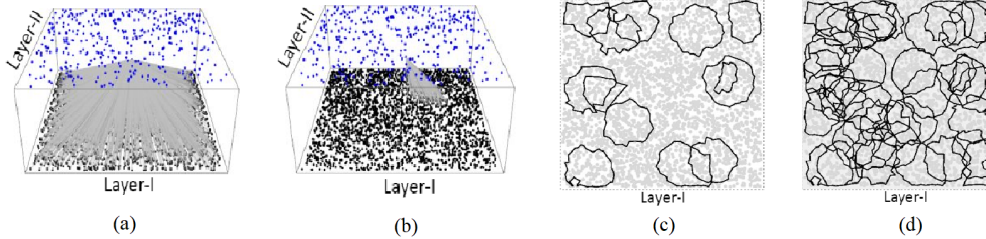

Figure 4: **Self-organization of Pooling layers.** (a) The initial configuration, wherein all nodes in the lower layer are connected to every unit in the higher layer. (b) After the self-organization process, a pooling architecture emerges, wherein every unit in layer-II is connected to a spatial patch of nodes in layer-I. (a,b) Here, connections from nodes in layer-I to a single unit in layer-II (higher layer) are shown. (c) Each contour represents a spatial patch of nodes in layer-I connected to a single unit in layer-II. (d) More than 95% of the nodes in layer-I are connected to units in the layer-II through well-defined pools, as the spatial patches tile layer-I completely.

## 4 Features of the developmental algorithm

In this section, we show that spatiotemporal waves can emerge and travel over layers with arbitrary geometries and even in the presence of defective sensor-nodes. As the local structure of sensor-node connectivity (local excitation and global inhibition) in the input layer in conserved over a broad range of macroscale geometries (figure 5a-d), we observe traveling activity waves on input layers with arbitrary geometries and in input-layers that have defects or holes. The coupling of the traveling activity wave in layer-I and a learning rule in layer-II results in the emergence of pooling architecture (refer to SI for an analytical treatment).

Furthermore, we demonstrate that the size and shape of the emergent spatiotemporal wave can be tuned by altering the topology of sensor-nodes in the layer. Coupling the emergent wave in layer-I with a learning rule in layer-II leads to localized receptive fields that tile the input layer.

Together, the wave and the learning rule endow the developmental algorithm with useful properties: (i) **Flexibility**: Spatial patches of sensor-nodes connected to units in layer-II can be established over arbitrary input-layer geometries. In Figure-5a, we show that an emergent spatiotemporal wave on a torus-shaped input layer coupled with the local learning rule (section-3.2) in layer-II, results in a pooling architecture. We also show that the developmental algorithm can self-organize networks on arbitrary curved surfaces (Figure-5b). Flexibility to form pooling layers on arbitrary input-layer geometries is useful for processing data acquired from unconventional sensors, like charge-coupled devices that mimic the retina [37]. The ability to self-organize pooling layers on curved surfaces makes it extremely useful for spherical image analysis. Spherical images acquired by omnidirectional cameras [38] placed on drones are becoming increasingly ubiquitous, and their analysis necessitates neural networks that can tile 3-dimensional surfaces. (ii) **Robustness**: Spatial patches of sensor-nodes connected to units in layer-II can be established in the presence of defective sensor nodes in layer-I. As shown in figure-5b, we initially self-organize a pooling architecture for a fully functioning set of sensor-nodes in the input-layer. To test robustness, we ablate a few sensor-nodes in the input-layer (captioned 'DN'). Following this perturbation, we observe that the pooling architecture re-emerges, wherein spatial-pools of sensor-nodes, barring the damaged ones, re-form and connect to units in layer-II. (iii) **Reconfigurable**: The size and shape of spatial pools generated can be modulated by tuning the structure of the emergent traveling wave (figure-5c & 5d). In figure-5e, we show that the size of spatial-pools can be altered in a controlled manner by modifying the topology of layer-I nodes. Wave-$x$ in the legend corresponds to an emergent wave generated in layer-I when every node in layer-I makes excitatory connections to other nodes in its 2 unit radius and inhibitory connections to every node above $x$ unit radius. This topological change alters the properties of the emergent wave, subsequently changing the resultant spatial-pool size. The histograms corresponding to these legends capture the distribution of spatial-pool sizes over all pools generated by a given wave-$x$. The histogram also highlights that the size of emergent spatial-pools are tightly regulated for every wave-configuration.

## 5 Growing a neural network

As the developmental algorithm (introduced in section 3) is flexible to varying scaffold geometries and tolerant to malfunctioning nodes, it can be implemented for growing a system, enabling us to push AI in the direction towards being more 'life-like' by reducing human involvement in the design of complex functioning architectures. The growth paradigm implemented in this section has been inspired by mechanisms that regulate neocortical development [39, 40].

The process of growing a layered neural network involves two major sub-processes. One, every 'node' can divide horizontally to produce daughter nodes that populates the same layer; two, every node can divide vertically to produce daughter processing units that migrate upwards to populate higher layers. Division is stochastic and is controlled by a set of random variables. Having defined the 3D scaffold, we seed a single unit (figure-6a). As horizontal and vertical division ensues to form the layered neural network, inter-layer connections are modified based on the emergent activity wave in layer-I and a learning rule (section-3.2) in layer-II, to form a pooling architecture. A detailed description of the growth rule-set coupled with a flow chart governing the growth of the network is appended to the supplemental materials.

Having intertwined the growth of the system and self-organization of inter-layer connections, we make the following observations: (1) spatiotemporal waves emerge in the first layer much before the entire layer is populated (figure-6b), (2) self-organization of inter-layer connections commences before the layered network is fully constructed (figure-6c) and (3) Over time, the system reaches a steady state as the number of 'cells' in the layered network remains constant and most processing units in the second layer connect to a pool of nodes in the first layer, resulting in the pooling architecture (figure-6d). Videos of networks growing on arbitrary scaffolds are added to the supplemental materials.

## 6 Growing functional neural networks

In the previous section, we demonstrated that we can successfully grow multi-layered pooling networks from a single unit. In this section, we show that these networks are functional.

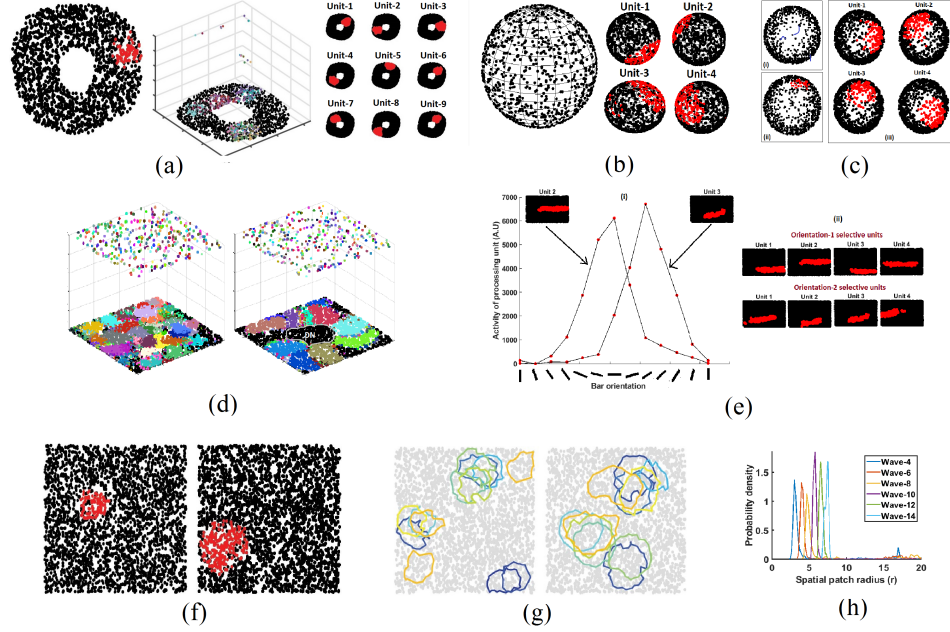

Figure 5: **Features of the developmental algorithm.** (a) **Self-organization of pooling layers for arbitrary input-layer geometry.** (a) The left most image is a snapshot of the traveling wave as it traverses layer-I; Layer-I has sensor-nodes arranged in an annulus geometry; red nodes refer to firing nodes. On coupling the spatiotemporal wave in layer-I to a learning rule in layer-II, a pooling architecture emerges. The central image refers to the 3d visualization of the pooling architecture, while each subplot in the right-most image depicts the spatial patch of nodes in layer-I connected to a single processing unit in layer-II. (b) Self-organizing pooling layers on a sphere. (b-ii) Upstream units connect to spatial patches of nodes on the sphere. (c) Self-organizing networks on Poincare disks with a hyperbolic distribution of input sensor nodes (c-ii) Snapshot of a traveling bump. (c-iii) Receptive fields of units in layer-II. (d) **Self-organization of pooling layers are robust to input layer defects** (d) The figure on the left depicts a self-organized pooling layer when all input nodes are functioning. Once these inter-layer connections are established, a small subset of nodes are damaged to assess if the pooling architecture can robustly re-form. The set of nodes within the grey boundary, titled 'DN', are defective nodes. The figure on the right corresponds to pooling layers that have adapted to the defects in the input layer, hence not receiving any input from the defective nodes.(e) (e-i) Tuning curve shows that units in layer-2 have a preferred orientation. (e-ii) Oriented receptive fields of units in layer-II. (f,g,h) **Pooling layers are reconfigurable.** (f) By altering layer-I topology (excitation/inhibition radii), we can tune the size of the emergent spatial wave. The size of the wave is 6 A.U (left) and 10 A.U (right). (g) Altering the size of the emergent spatial wave tunes the emergent pooling architecture. The size of the pools obtained are 4 A.U (left), obtained from a wave-size of 6 A.U and a pool-size of 7 A.U (right), obtained from a wave-size of 10 A.U. (h) A large set of spatial-pools are generated for every size-configuration of the emergent wave. The distribution of spatial-pool sizes over all pools generated by a specific wave-size are captured by a kernel-smoothed histogram. Wave-4 in the legend corresponds to a histogram of pool-sizes generated by an emergent wave of size 4 A.U (blue line). We observe that spatial patches that emerge for every configuration of the wave have a tightly regulated size.

We demonstrate functionality of networks grown and self-organized from a single unit (figure-7c) by evaluating their train and test accuracy on a classification task. Here, we train networks to classify images of handwritten digits obtained from the MNIST dataset (figure-7e). To interpret the results, we compare it with the train/test accuracy of hand-crafted pooling networks and random networks. Hand-crafted pooling networks have a user-defined pool size for all units in layer-II (figure-7b), while random networks have units in layer-II that connect to a random set of nodes in layer-I without any spatial bias (figure-7d), effectively not forming a pooling layer.

To test functionality of these networks, we couple the two-layered network with a linear classifier that is trained to classify hand-written digits from MNIST on the basis of the representation provided

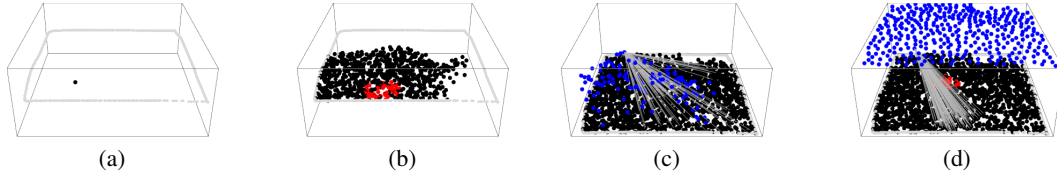

<center>(a)            (b)            (c)            (d)</center>

Figure 6: **Growing a layered neural network** (a) A single computational "cell" (black node) is seeded in a scaffold defined by the grey boundary. (b) Once this "cell" divides, daughter cells make local-excitatory and global-inhibitory connections. As the division process continues, noisy interactions between nodes results in emergent spatiotemporal waves (red-nodes). (c) Some nodes within layer-I divide to produce daughter cells that migrate upwards to form processing units (blue nodes). The connections between the two layers are captured by the lines that connect a single unit in a higher layer to nodes in the first layer (Only connections from a single unit are shown).(d) After a long duration, the system reaches a steady state, where two layers have been created with an emergent pooling architecture.

by these three architectures (hand-crafted, self-organized and random networks). We observe that self-organized networks classify with a 90% test accuracy, are statistically similar to hand-crafted pooling networks (90.5%, p-value = 0.1591) and are statistically better than random networks (88%, p-value = 5.6 x $10^{-5}$) (figure-7a). Performance is consistent over multiple self-organized networks. These results demonstrate that self-organized neural networks are functional and can be adapted to perform conventional machine-learning tasks, with the additional advantage of being autonomously grown from a single unit.

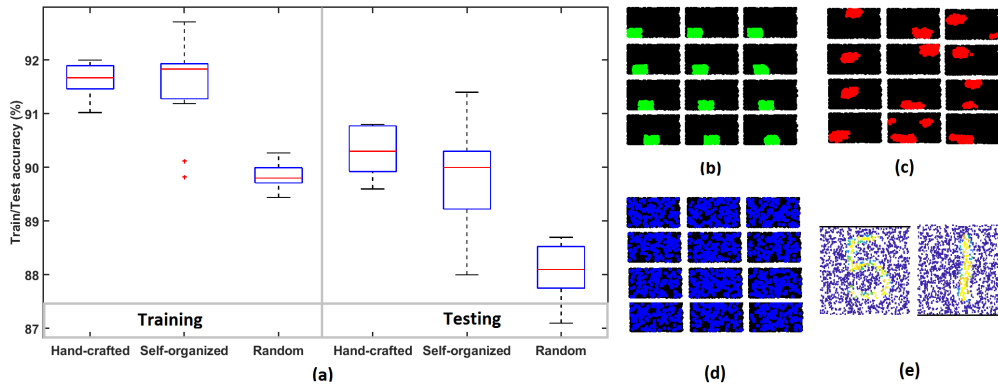

Figure 7: **Networks grown from a single unit are functional.** Three kinds of networks are trained and tested on images obtained from the MNIST database. We use 10000 training samples and 1000 testing samples. The 3 kinds of networks are: (i) Hand-crafted, (ii) Self-organized networks and (iii) random networks. The training procedure is run over n=11 networks to ensure that the developmental algorithm always produces functional networks. (a) The box-plot captures the training and testing accuracy of these 3 networks. We notice that the testing accuracy of self-organized networks is comparable to that of to that of hand-crafted networks (p-value = 0.1591>0.05) and are much better than random networks (p-value = 5.6 x $10^{-5}$). (b,c,d) Each unit in the second layer is connected to a set of nodes in the lower layer. The set it is connected to are defined by the green, red or blue nodes in the subplots shown. (b) Hand-crafted (c) Self-organized and (d) Random-basis.(e) Two MNIST images as seen in the first layer.

## 7 Discussion

In this paper, we address a pertinent question of how artificial computational machines could be built autonomously with limited human intervention. Currently, architectures of most artificial systems are obtained through heuristics and hours of painstaking parameter tweaking. Inspired by the development of the brain, we have implemented a developmental algorithm that enables the robust growth and self-organization of functional layered neural networks.

Implementation of the growth and self-organization framework brought many crucial questions concerning neural development to our attention. Neural development is classically defined and abstracted as occurring through discrete steps, one proceeding the other. However in reality, development is a continuous flow of events with multiple intertwined processes [41]. In our work on growing artificial systems, we observed the mixing of processes that control growth of nodes and self-organization of connections between nodes. The mixing of growth and connection processes got us interested in how timing can be controlled when processes occur in parallel.

The work also reinforces the significance of brain-inspired mechanisms for initializing functional architecture to achieve generalization for multiple tasks. A peculiar instance in the animal kingdom is the presence of precocial species, animals whose young are functional immediately after they are born (examples include domestic chickens, horses) . One mechanism that enables functionality immediately after birth is spontaneous activity that assists in maturing neural circuits much before the animal receives any sensory input. Although we have shown how a layered architecture (mini-cortex) can emerge through spontaneous activity in this paper, our future work will focus on growing multiple components of the brain, namely a hippocampus and a cerebellum, followed by wiring these regions in a manner useful for an organism's functioning. This paradigm of growing mini-brains in-silico will allow us to (i) explore how different components in a biological brain interact with one another and guide our design of neuroscience experiments and (ii) equip us with systems that can autonomously grow, function and interact with the environment in a more 'life-like' manner.

**Acknowledgments**

We would like to thank Markus Meister, Carlos Lois, Erik Winfree, Naama Barkai for their invaluable contribution for shaping the early stages of the work. We also thank Alex Farhang, Jerry Wang, Tony Zhang, Matt Rosenberg, David Brown, Ben Hosheit, Varun Wadia, Gautam Goel, Adrianne Zhong and Nasim Rahaman for their constructive feedback and key edits that have helped shape this paper.

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
