[Supplementary Material]

# Appendix

## 1 Mathematical model

### 1.1 Dynamical model for input sensor nodes

Input sensor nodes are modeled using the Izhikevich neuron model. We selected the Izhikevich model primarily because it has the least number of parameters for accurately modeling neuron-like activity and the parameter regimes that produce different neuronal firing states have been well characterized earlier [1].

---

**Dynamical model for input-sensor nodes in the lower layer (layer-I):**

$$\frac{\mathrm{d}v_i}{\mathrm{d}t} = 0.04v_i^2 + 5v_i + 140 - u_i + \sum_{j=1}^{N} S_{i,j}\mathcal{H}(v_j - 30) + \eta_i(t) \tag{1}$$

$$\frac{\mathrm{d}u_i}{\mathrm{d}t} = a_i(b_i v_i - u_i) \tag{2}$$

with the auxiliary after-spike reset:

$$v_i(t) > 30, \text{then}: \begin{cases} v_i(t + \Delta t) = c_i \\ u_i(t + \Delta t) = u_i(t) + d_i \end{cases}$$

where: (1) $v_i$ is the activity of sensor node $i$; (2) $u_i$ captures the recovery of sensor node $i$; (3) $S_{i,j}$ is the connection weight between sensor-nodes $i$ and $j$; (4) $N$ is the number of sensor-nodes in layer-I; (5) Parameters $a_i$ and $b_i$ are set to 0.02 and 0.2 respectively, while $c_i$ and $d_i$ are sampled from the distributions $\mathcal{U}(-65, -50)$ and $\mathcal{U}(2, 8)$ respectively. Once set for every node, they remain constant during the process of self-organization. The initial values for $v_i(0)$ and $u_i(0)$ are set to -65 and -13 respectively for all nodes. These values are taken from Izhikevich's neuron model [1]; (6) $\eta_i(t)$ models the noisy behavior of every node $i$ in the system, where $< \eta_i(t)\eta_j(t') > = \sigma^2 \, \delta_{i,j}\delta(t - t')$. Here, $\delta_{i,j}$, $\delta(t - t')$ are Kronecker-delta and Dirac-delta functions respectively, and $\sigma^2 = 9$; (7) $\mathcal{H}$ is the unit step function:

$$\mathcal{H}(v_i - 30) = \begin{cases} 1, & v_i \geq 30 \\ 0, & v_i < 30. \end{cases}$$

---

### 1.2 Topology of input-sensor nodes

The nodes in the lower layer (layer-I) are arranged in a local-excitation, global inhibition topology, with a ring of nodes between the excitation and inhibition regions that have neither excitation or inhibition (zero weights) . We have observed that the zero-weight ring that has no connections between the excitation and inhibition regions gives us a good control over the emergent wave size. This is detailed in Box-1.2 and depicted in figure-1a.

NETWORK TOPOLOGY

NODE OF INTEREST
LOCAL EXCITATION
NO CONNECTION
GLOBAL INHIBITION

EACH NODE HAS THE
SAME NETWORK TOPOLOGY

LAYER I (input-layer)

(a)

(b)

Figure 1: Topology of sensor-node connections: Every node is connected to other nodes in the layer within a radius $r_e$ via a positive weight, not connected to nodes positioned at a distance between $r_e$ and $r_i$ and connected to nodes at a distance larger than $r_i$ with a decaying negative weight.

## 1.3 Modeling Processing units and winner-take-all strategy

Processing units are modeled as Rectified linear units (ReLU) associated with an arbitrary threshold. Although the threshold is randomly initialized, it is updated during the process of self-organization. Threshold update depends entirely on the activity trace of the associated processing unit . We also require that, at every time point, at most a single processing unit in layer-II be activated by the emergent patterned activity in layer-I. To enforce single layer-II unit firing, we let the processing units, modeled as ReLU units compete with each other in a winner-take-all (WTA) manner. WTA dynamics ensures that at every time point, at most a single unit in layer-II responds to the patterned activity in the input layer.

Each processing unit in layer-II is modeled by the equation given below:

$$y_j(t) = \mathcal{W}[\max(0, \sum_{i=1}^{N} w_{i,j}(t)\mathcal{H}(v_i(t) - 30))] \tag{3}$$

Here, the $\max(0, x)$ is the implementation of a rectified linear unit (ReLU); $\mathcal{H}(v_i(t) - 30)$ is the threshold activity of sensor node $i$ (in layer-I) at time 't'; $y_j(t)$ is the activation of processing unit $j$ (in layer-II) at time 't'; $w_{i,j}^t$ is the connection weight between sensor-node $i$ and processing unit $j$ at

time 't'; $N$ is the number of sensor-nodes in layer-I and $\mathcal{W}$ refers to the winner-take-all mechanism that ensures a single winning processing unit.

The winner-take-all function implemented in layer-II is mathematically elaborated below:

$$\mathcal{W}[y_j(t)] = \begin{cases} \max(0, y_j(t) - c_j(t)), & \text{if } y_j(t) > y_k(t) \quad \forall k \in [1,...j-1,j+1,...,M] \\ 0 & \text{otherwise} \end{cases}$$

Here, $y_j(t)$ is the activation of processing unit $j$ (in layer-II) at time 't'; $c_j(t)$ is the threshold for processing unit $j$ at time 't' and $M$ is the number of processing units in layer-II. Every processing unit is modeled as a ReLU with an associated threshold ($c_j$). Although this threshold is arbitrarily initialized, they are updated during the process of self-organization. The update depends on the number of times the connections between processing units and nodes in layer-I are updated as described below.

To implement threshold update, we keep track of the number of times connections between a specific processing unit and sensor nodes in layer-I are updated over the course of 1000 time-points. $z_j(t)$ captures the number of times connections between processing unit-j and sensor-nodes in layer-I are updated.

**Recording the synaptic changes per processing unit:**

$$z_j(t+1) = \begin{cases} z_j(t) + 1 & \text{if } (y_j(t) > 0) \\ 0 & \text{if } (t \bmod 1000) = 0 \\ z_j(t) & \text{otherwise} \end{cases}$$

The threshold for a processing unit is updated based on the number of connections that were altered in the past 1000 time points between that processing unit and sensor-nodes in layer-I.

**Updating the threshold for every processing unit:**

$$c_j(t+1) = \begin{cases} \max(y_j(t), y_j(t-1), ..., y_j(0))/5, & \begin{array}{l} \text{if } (t \bmod 1000) = 0 \text{ AND} \\ z_j(t) < 200 \end{array} \\ c_j(t) & \text{otherwise} \end{cases}$$

Here, $w_{i,j}(t)$ is the weight of connection between sensor-node $i$ and processing unit $j$ at time 't'; $\eta_{learn}$ is the learning rate; $y_j^t$ is the activation of processing unit $j$ at time 't'; $z_j(t)$ is the number of synaptic modifications made to unit $j$ until time 't'; ($t \bmod 1000$) is the remainder when $t$ is divided by 1000 and $c_j(t)$ is the activation threshold for processing unit $j$ at time 't'.

The emergent wave in layer-I coupled with the learning rule implemented by processing units in layer-II are sufficient to self-organize pooling architectures.

## 2  Growing a neural network

We demonstrate that by defining a minimal set of 'rules' for a single computational 'cell', we can grow a layered network, followed by the self-organization of its inter-layer connections to form pooling layers.

In order to grow a layered network, we define a 3D scaffold and seed the first layer in the scaffold with a computational 'cell' (figure-2a). The major attributes of nodes in the first layer are:

- $v_i(t)$ : activity of node $i$ modeled by the Izhikevich equation [1]
- $clockH_i$ : records the age of the 'cell', allowing horizontal division (division within the same layer) until it reaches a certain age
- $HFlim_i$ : the maximum divisions permitted for node $i$
- $VCD_i$ : a binary variable that records whether node $i$ has vertically divided or not. Vertical division is the process when a 'cell' divides and its daughter 'cells' migrate upwards to form processing units that populate higher layers.

Figure 2: **Growing a layered neural network** (a) A single computational "cell" (black node) is seeded in a scaffold defined by the grey boundary. (b) Once this "cell" divides, daughter cells make local-excitatory and global-inhibitory connections. As the division process continues, noisy interactions between nodes results in emergent spatiotemporal waves (red-nodes). (c) Some nodes within layer-I divide to produce daughter cells that migrate upwards to form processing units (blue nodes). The connections between the two layers are captured by the lines that connect a single unit in a higher layer to nodes in the first layer (Only connections from a single unit are shown).(d) After a long duration, the system reaches a steady state, where two layers have been created with an emergent pooling architecture.

## 2.1 User-defined Growth Parameters

| Parameter | Value | Description |
|---|---|---|
| HCD_AGE | 25 | The maximum time a cell can pursue horizontal division |
| HF_MAX | 40 | The maximum number of divisions a single cell can pursue |
| R_HDIV | 1 | Critical radius I |
| R_VDIV | 1 | Critical radius II |
| THRESH_HDIV | 3 | The maximum number of cells permitted within a radius (R_HDIV) |

## 2.2 Growth Process

**Step: 1**:

A single computational 'cell' endowed with the following attributes is seeded on a 3D scaffold. The attributes and values that a seeded computational 'cell' is endowed with is mentioned in the table below. The first column indicates attributes, second column denotes the initial values that they take and the third column is a description of the attribute.

| Cell attribute | Initialization | Description |
|---|---|---|
| $v_i$ | -65 | Initialize activity of node $i$ |
| $clockH_i$ | 0 | Initializing clock to 0, for every newly divided daughter cell |
| $HFlim_i$ | HF_MAX | Initializing the max divisions to HF_MAX for the seeded cell. |
| $VCD_i$ | 0 | Before vertical division, $VCD_i = 0$; After vertical division, $VCD_i = 1$; |

### 2.2.1 Step: t → t+1

A random cell $i$ is sampled from the input layer.

If the cell hasn't crossed the critical age threshold ($clockH_i <$ HCD_AGE) and the number of cells within a radius (R_HDIV) is below the density threshold ($numCells_i$(R_HDIV) $<$ THRESH_HDIV), the cell divides horizontally to form daughter cells that populate the same layer. The clockH is reset to zero for the daughter cells, however the HFlim attribute of the daughter cells is one less than their parent to keep track of the number of divisions.

If it hasn't reached the critical age threshold, but has a local density above the defined density threshold, it remains quiescent and a new 'cell' is sampled.

A cell $i$ can divide vertically only if the cell has reached the critical age threshold ($clockH_i$ = HCD_AGE) and cells in its local vicinity (with radius :- R_VDIV) haven't divided vertically. As

mentioned in an earlier section, a binary variable $VCD_i$ keeps track of whether a cell has divided vertically or not.

When a cell divides vertically, one daughter cell occupies the parent's position on layer-I, while the other daughter cell migrates upwards. The daughter cell that migrates upwards initially makes a single connection with its twin on layer-I, which gets modified with time, resulting in a pool of nodes in layer-I making connections with a single unit in the higher layer (pooling architecture).

### 2.2.2 Termination condition

The local rules that control horizontal division and vertical division are active throughout and prevent the system from blowing up, with respect to the number of nodes in each layer. It has been observed that the system reaches a steady state, as the number of 'cells' in both layers remain constant.

Figure 3: Growth flowchart

### 2.3 Growing neural networks on arbitrary scaffolds (Results)

Videos of multi-layered networks growing on arbitrary scaffolds can be viewed by visiting this link: [https://drive.google.com/open?id=1YtFEvWHTU9HWl760V8lEr9Heapx0sUdh]

## 3 Minimal model for observing emergent spatiotemporal waves

In this section, we provide an analytical solution for the emergence of a spatiotemporal wave through noisy interactions between constituent nodes in the same layer.

As we stated in the main-text, the key ingredients for having a layer of nodes function as a spatiotemporal wave generator are:

- Each sensor-node should be modeled as a dynamical systems model
- Sensor-nodes should be connected in a suitable topology (here, local excitation ($r_e < 2$ and global inhibition ($r_i > 4$).

On modeling all nodes in the system using a simple set of ODE's, we highlight the conditions required for observing a stationary bump in a network of spiking sensor-nodes and to observe instability of the stationary bump resulting in a traveling wave.

## 3.1 Arranging sensor-nodes in a line

We choose a configuration where $N$ sensor-nodes are randomly arranged in a line (as shown in figure-4).

Figure 4: Sensor nodes arranged in a line

The activity of $N$ sensor nodes, arranged in a line as in figure-4, are modeled using a minimal ODE model as described below:

$$\tau_d \frac{\mathrm{d}x(u_i,t)}{\mathrm{d}t} = -x(u_i,t) + \sum_{u_j \in \mathcal{U}} S(u_i,u_j)\mathcal{F}(x(u_j,t)) \quad \forall i \in 1,...,N \tag{4}$$

Here, $u_i$ represents the position of nodes on a line; $x(u_i,t)$ defines the activity of sensor node positioned at $u_i$ at time $t$; $S_{u_i,u_j}$ is the strength of connection between nodes positioned at $u_i$ and $u_j$; $\tau_d$ controls the rate of decay of activity; $\mathcal{U}$ is the set of all sensor nodes in the system $(u_1,u_2,...,u_N)$ for $N$ sensor nodes; and $\mathcal{F}$ is the non-linear function required to convert activity of nodes to spiking activity. Here, $\mathcal{F}$ is the heaviside function with a step transition at 0.

Each sensor-node has the same topology of connections, ie fixed strength of positive connections between nodes within a radius $r_e$, no connections from a radius $r_e$ to $r_i$, and decaying inhibition above a radius $r_i$. This is depicted in figure-5

Figure 5: strength of connections between sensor-nodes

### 3.1.1 Fixed point analysis

We determine the stable activity states of nodes placed in a line by a fixed point analysis, similar to what Amari developed in [2] for the case when there are infinite nodes.

$$x(u_i) = \sum_{u_j \in \mathcal{U}} S(u_i,u_j)\mathcal{F}(x(u_j)) \quad \forall i \in 1,...,N \tag{5}$$

On solving this system of non-linear equations simultaneously, we get a fixed point ie a vector $x^*$ $\in \mathcal{R}^N$, corresponding to the activity of $N$ sensor nodes positioned at $(u_1,u_2,...,u_N)$. To assess their spiking from the activity of sensor-nodes, we have

$$s_i = \mathcal{F}(x(u_i)) \quad \forall i \in 1,...,N \tag{6}$$

As the weight matrix $(S_{u_i,u_j})$ used incorporates the local excitation $(r_e < 2)$ and global inhibition $(r_i > 4)$ ( figure-5), we get solutions with a single bump of activity (figure-6a), two bumps of activity (figure-6c) or a state when all nodes are active.

|     |     |     |
| --- | --- | --- |
| (a) | (b) | (c) |

Figure 6: **Fixed points:** Multiple fixed points are obtained by solving $N$ non-linear equations simultaneously. Some of the solutions obtained are: (a) a single bump at the center, (b) a single bump at one of the edges and (c) two bumps of activity.

### 3.1.2  Stability of fixed points

To assess the stability of these fixed points, we evaluate the eigenvalues of the Jacobian for this system of differential equations. As there are $N$ differential equations, the Jacobian ($\mathbb{J}$) is an $N$x$N$ matrix.

$$\frac{\mathrm{d}x(u_i,t)}{\mathrm{d}t} = \frac{-x(u_i,t)}{\tau_d} + \sum_{u_j \in \mathcal{U}} \frac{S(u_i,u_j)\mathcal{F}(x(u_j))}{\tau_d}$$

$$\frac{\mathrm{d}x(u_i,t)}{\mathrm{d}t} = f_i(u_1,u_2,...,u_N)$$

$$f_i(u_1,u_2,...,u_N) = \frac{-x(u_i)}{\tau_d} + \sum_{u_j \in \mathcal{U}} \frac{S(u_i,u_j)\mathcal{F}(x(u_j))}{\tau_d} \tag{7}$$

$$\mathbb{J}(i,j) = \frac{\partial f_i(u_1,u_2,...,u_N)}{\partial x(u_j)}$$

On evaluating the Jacobian ($\mathbb{J}$) at the fixed points obtained ($x^*$), we get:

$$\mathbb{J}(i,i) = \frac{\partial f_i}{\partial x(u_i)}$$

$$\mathbb{J}(i,i) = \frac{-1}{\tau_d}$$

$$\mathbb{J}(i,j) = S(u_i,u_j)\mathcal{F}^{'}(x(u_j))\frac{\partial x(u_j)}{x(u_j)} \tag{8}$$

$$\mathbb{J}(i,j) = S(u_i,u_j)\delta(x(u_j))$$

$$\mathbb{J}(i,j) = 0 \;\; \forall x(u_j) \neq 0$$

Here, $\mathcal{F}$ is the Heaviside function and its derivative is the dirac-delta($\delta$); where, $\delta(x) = 0$, for $x \neq 0$ and $\delta(x) = \infty$ for $x = 0$.

For a fixed point, where $x^*(u_k) \neq 0$, $\forall k \in 1,...,N$, the Jacobian is a diagonal matrix with $\frac{-1}{\tau_d}$ in its diagonals. This implies that the eigenvalues of the Jacobian are $\frac{-1}{\tau_d}$ ($\tau_d > 0$), which assures that the fixed point $x* \in \mathcal{R}^N$ is a stable fixed point.

### 3.1.3  Destabilizing the fixed point

With the addition of high amplitude of gaussian noise to the ODE's described earlier, we can effectively destabilize the fixed point, resulting in a traveling wave. The equations with the addition of a noise term are:

$$\tau_d\frac{\mathrm{d}x(u_i,t)}{\mathrm{d}t} = -x(u_i,t) + \sum_{u_j \in \mathcal{U}} S(u_i,u_j)\mathcal{F}(x(u_j,t)) + \eta_i(t) \;\; \forall i \in 1,...,N \tag{9}$$

Here, $\eta_i(t)$ models the noisy behavior of every node $i$ in the system, where $< \eta_i(t)\eta_j(t') > = \sigma^2 \delta_{i,j}\delta(t-t')$. Here, $\delta_{i,j}$, $\delta(t-t')$ are Kronecker-delta and Dirac-delta functions respectively, and $\sigma^2$ captures the magnitude of noise added to the system.

The network of sensor nodes is robust to a small amplitude of noise ($\sigma^2 \in (0,4)$), while a larger amplitude of noise ($\sigma^2 > 5$) can destabilize the bump, forcing the system to transition to another bump

in its local vicinity. Continuous addition of high amplitudes of noise forces the bump to move around in the form of traveling waves. The behavior is consistent with the linear stability analysis because noise can push the dynamical system beyond the envelop of stability for a given fixed point solution.

## 3.2 Arranging sensor nodes in a 2D square

In this section, we arrange $N$ sensor nodes arbitrarily on a 2-dimensional square as shown in figure-7, with the same local structure (local excitation and global inhibition).

The activity of these sensor nodes are modeled using the minimal ODE model described earlier (in equation-4).

Figure 7: Sensor nodes placed arbitrarily on a square plane

We obtain the fixed points ($x^* \in \mathcal{R}^N$), by solving $N$ simultaneous non-linear equations using BBsolve [3]. We notice that the fixed point solutions have a variable number of activity bumps in the 2D plane as shown in figure-8a,8b & 8c.

| (a) | (b) | (c) |

Figure 8: **Stable Fixed points:** Multiple fixed points are obtained by solving $N$ non-linear equations simultaneously. Some of the solutions obtained are: (a) a single bump, (b) two bumps and (c) three bumps of activity.

## 3.3 Arranging sensor nodes on a 2D sheet of arbitrary geometry

In this section, we arrange sensor nodes on a 2D sheet in any arbitrary geometry as shown in figure 9. Although the macroscopic geometry of the sheet changes, the local structure of sensor nodes in conserved (ie local excitation and global inhibition).

The fixed points are evaluated by simultaneously solving the non-linear system of equations. We notice that the bumps are stable fixed points even when sensor nodes are placed on a 2-dim sheet of arbitrary geometry.

| (a) | (b) | (c) | (d) |

Figure 9: **Stable Fixed points:** Multiple fixed points are obtained by solving $N$ non-linear equations simultaneously. Some of the solutions obtained are: (a,b) a single bump for a circular geometry (c,d) two bumps of activity for arbitrary geometry

# 4 Growing functional neural networks

We estimate functionality of networks grown and self-organized from a single unit by evaluating their train and test accuracy on a classification task. Here, we train networks to classify images of handwritten digits obtained from the MNIST dataset. To interpret the results, we compare it with the train/test accuracy of hand-crafted pooling networks and random networks. Hand-crafted pooling networks have a user-defined pool size for all units in layer-II, while random networks have units in layer-II that connect to a random set of nodes in layer-I without any spatial bias, effectively not forming a pooling layer.

To test functionality of these networks, we couple the two-layered network with a linear classifier that is trained to classify hand-written digits from MNIST on the basis of the representation provided by these three architectures (hand-crafted, self-organized and random networks).

The first two layers in the network serve as feature extractors, while the last layer behaves like a perceptron. The optimal classifier is learnt by minimizing the least square error between the output of the network and a desired target. However, there isn't any back-propagation through the entire network. In essence, the architecture grown through the developmental algorithm remains fixed, performing the task of latent feature representation, while the classifier learns how to match these latent features with a set of task-based labels.

## 4.1 Setting up the pooling architecture

The first two layers of the network correspond to the pooling architecture grown by the developmental algorithm. The input is fed to the first layer, while the units in the second layer, that are connected to spatial pools in layer-I, extract features from these inputs.

Let $x \in \mathcal{R}^N$ be the input data (for $N$ sensor nodes) and the weights connecting the first and second layer be $W_1 \in \mathcal{R}^{M \times N}$ (for $M$ processing units). The features extracted in layer-II are: $y = \mathcal{F}(W_1 x)$. Here, $\mathcal{F}$ is any non-linear function applied to the transformation in order to map all the values in layer-II within the range [-1,1].

## 4.2 Appending a fully connected layer

The pooling architecture sends its feature map through a fully connected layer with $L$ nodes, with the weights connecting the set of processing units and the fully connected layer being randomly initialized as $W_2 \in \mathcal{R}^{L \times M}$. The features extracted by the fully connected layer are: $y_{FC} = \mathcal{F}(Wy)$. $\mathcal{F}$ is the same as the one used in section-4.1.

## 4.3 Classification accuracy

The final set of weights connecting the fully connected layer to the 10 element vector (as there are 10 digit classes in the MNIST dataset) is denoted by $W_3 \in \mathcal{R}^{10 \times L}$. The output generated by the network is $y_O = W_3 y_{FC}$. Let us denote the target output as $y_T$.

As we want to minimize the least square error between the target output ($y_T$) and output of the network ($y_O$), conventionally, we can perform a gradient descent. However, as it is a linear classifier, we have a closed form solution for the weight matrix ($W_3$).

$$y_O = W_3 y_{FC}$$
$$y_T = W_3 y_{FC} \qquad \text{for zero error, } y_0 = y_T$$
$$y_T y_{FC}^T = W_3 y_{FC} y_{FC}^T$$
$$W_3 = y_T y_{FC}^T (y_{FC} y_{FC}^T)$$

Setting the weights between the fully connected layer and the output layer ($W_3 = y_T y_{FC}^T (y_{FC} y_{FC}^T)$), we evaluate the train and test accuracy for 3 kinds of networks. (Hand-crafted pooling, self-organized and random networks). These networks differ primarily in how their first two layers are connected. The hand-programmed pooling networks are those that have a fixed size of spatial pool that connects to units in layer-II, while the random networks have no spatial pooling.

The results are described in the main-paper and we observe that self-organized networks classify with a 90% test accuracy are statistically similar to hand-crafted pooling networks (90.5%, p-value = 0.1591) and statistically better than random networks (88%, p-value = 5.6 x $10^{-5}$) (figure-7a). This performance is consistent over multiple self-organized networks. The train/test accuracy of self-organization networks highlights that growing networks through a brain-inspired developmental algorithm is potentially useful to building functional networks.

## 5 Scalability: Determining the speed of self-organization of the pooling architecture as the size of the input-layer increases

Here, we demonstrate that the pooling layers can be self-organized for very large input layers. Large layers are defined based on the number of sensor nodes in the layer. We observe that enforcing a spatial bias on the initial set of connections from units in layer-II to the nodes in the input layer, enables us to speed up the process of self-organization.

Our simulations show that the self-organization of pooling layers can be scaled up to large layers (with upto 50000 nodes) without being very expensive, as an increase in number of sensor-nodes results in multiple simultaneous waves tiling the input layer, effectively forming a pooling architecture in parallel.

(a) Input layer: 1500 nodes     (b) Input layer: 5000 nodes     (c) Input layer: 10000 nodes

(d) Time complexity for self-organization of pooling layers

Figure 10: **Developmental algorithm scales efficiently to very large input layers:** (a) Layer-I has 1500 nodes and layer-II has 400 nodes. The emergent wave in layer-I results in a single traveling wave that tiles layer-I. (b) Layer-I has 5000 nodes and layer-II has 400 nodes. The emergent wave in layer-I results in a single traveling wave that tiles layer-I. (c) Layer-I has 10000 nodes and layer-II has 400 nodes. The emergent wave in layer-I results in a multiple traveling wave that tile layer-I simultaneously. This results in a single processing unit receiving pools from different regions. (d) The histogram captures the time taken for a pooling layer to form for variable number of input sensor nodes (1500, 5000, 10000, 25000 and 50000 nodes). With an increase in the number of sensor-nodes, the speed of self-organization increases as multiple waves tile the input layer simultaneously.