[Reviews · NeurIPS 2019]

Reviewer 1



- It seems like the second layer in the model is lower dimensional than the first layer. Is there evidence of a dimensionality reduction from retina and LGN that would match this feature of the model? - "most artificial systems are obtained through heuristics and hours of painstaking parameter tweaking." => Does not sound like a relevant comparison, because these artificial systems can solve much more complicated tasks than MNIST. Clarity: - The task on which the network was tested (MNIST) should be mentioned in the abstract. - "The algorithm organizes inter-layer connections to construct a convolutional pooling layer, a key constituent of convolutional neural networks" => The term "convolutional" implies weight tying, but here you can only obtain locally connected units, without weight-tying. Please change language here and everywhere CNNs are mentioned. "Our work on growing artificial systems got us interested in how critical times of different developmental processes are controlled, and whether they were controlled by an internal clock." => Please share your thoughts on this interesting aspect (or remove this sentence). Originality: looks original, but the authors you be more explicit about how you work is different than refs [25->29]. In particular, can you expand on what these other studies contributed? Significance: Interesting results for neuroscience. How could this network apply to ML (beyond a slight benefit on MNIST compared to random networks)? Maybe new bio-inspired hardware?

Reviewer 2



The main contributions of this paper are to propose an algorithm to learn a pooling architecture and one to grow the architecture with only self-organization principles. The developmental algorithm is evaluated on a different input geometry and on experiments with faults in the first layer. A last experiment evaluates the proposed algorithms on a MNIST classification task. I like the originality of the work, as the authors propose the principle of a growing machine, that is able to yield a functional architecture from a limited set of rules. The principles to follow for building such self-organized network are clearly exposed. The concerns expressed below have been answered by the authors' rebuttal, I appreciate that network is implemented with spiking neurons relying on bio-inspired hardware, as it has nice properties for processing temporal input stream. The results could be more convincing with a task relying on coincidence detection or time-structured events, like audio processing, echolocation or event-based vision. The authors argued in the rebuttal that the global inhibition scheme is limited in space, indicating that it does not scale with the size of the network which was my main concern. This is thus a more biologically feasible approach that is compatible with hardware implementation. The "flexibility" of the network allows the network to address the so-called packing problem, that is how to efficiently cover the sensor space. The authors shown in their response convincing experimental results with a hyperbolic geometry that have non-uniform density. -- Original comments -- I could not understand why the authors rely on a spiking model for the layer-I and ReLU units for layer-II. Model of sensory inputs with intrinsic noise could be modeled with neural masses or discrete neural fields. What is the benefit of spiking neurons in this contribution? On the layer-I, the local-excitation and global-inhibition scheme is encountered in the literature but it is limited by the inhibition range. Biological observations are seldom in favor of a global inhibition, except in organisms with a limited brain size (mainly insects) as a truly global inhibition for all neurons require a complex network of connections. In my opinion, the proposed self-organization for arbitrary input-layer geometry of Sect. 4 is not convincing enough, mainly because the tiling of the input layer is uniform. I think that variation in the density of input sensors could better demonstrate the approach, for example following with hyperbolic distribution (like Poincaré Disk). In Sect. 6, I find that the random networks are performing very well on MNIST classification, how a network with random connectivity fed by spiking neurons is performing around 88% accuracy? As a small note, I did not find the movie of Fig. 2 in the supplementary material. This approach is highly original and clearly exposed. The quality and significance are less clear as this approach is difficult to compare to existing methods.

Reviewer 3



[Update after author's reply:] The authors have produced a strong response, with important additional results. I am switching my assessment to "moderate accept", UNDER the condition that the authors include these new results in the paper AND replace occurrences of "convolutional" with "retinotopic". [Original review below:] First, the author oversell their results considerably, claiming to observe the emergence of "convolutional" and "pooling" cells in the sense of convolutional neural networks, which would indeed be an interesting result. However, the learned receptive fields are not "convolutional": there is no feature-specific filter being replicated across the map. Indeed, there isn't any feature selectivity at all! They cannot be called "pooling" either, unless the term is extended to apply to any receptive field whatsoever. The proper term that the author are looking for is "retinotopy" - the neurons learn to restrict their inputs to a specific topological neighbourhood, i.e. they become retinotopic. This is considerably below the state of the art from the 90s, when various authors demonstrated the emergence of feature-selective neurons, with orientation maps and higher-level selectivity. I encourage the authors to consult the work of Poggio (Riesenhuber and Poggio, Serre and Poggio, Masquelier and Poggio) and Rolls (especially Treves & Rolls, Stringer & Rolls). Miikkulainen is cited, but not for his work on the highly relevant LISSOM map-learning system. All of these systems showed selectivity for features of varying complexity - certainly much more complex than pure spatial location (i.e. retinotopy) as shown here. (Note: I am not affiliated with any of these authors) The text is otherwise clear and well-written and seems technically sound. Originality: 1/5 Quality: 4/5 Clarity: 4/5 Significant: 1/5 My overall score is 4 rather than 3, in order to encourage the authors to work more on this project and improve it for future re-submission (possibly at a workshop). See "Improvements".

[Author Response · NeurIPS 2019]

We would like to thank the reviewers for their feedback. We are glad that each of the reviewers had positive comments about the submission. R1 stated that the paper is "original" and has "interesting results for neuroscience". R2 "liked the originality of the work" and found the algorithm "clearly exposed". R3 expressed that "the paper is clear, well-written and technically sound".

The reviewers wanted to better understand the utility of the developmental strategy over traditional CNN's. As one example, CNN's are primarily used for analyzing 2d planar images. However, there is a growing need to analyze spherical images acquired by omnidirectional cameras on cars, drones (ref: omnidirectional camera- Davide Scaramuzza, UPenn). A 2d projection of a spherical image distorts the image, necessitating that a neural-net tile 3d curved surfaces, a challenge for hand-crafted CNNs. Our developmental algorithm, unlike traditional CNNs, self-organizes based on local rules, and can form pooling layers that tile curved surfaces. Here, we show the self-organization of a pooling layer on a sphere (fig-1a).

Figure 1: (a) Self-organizing pooling layers on a sphere. (a-ii) Upstream units connect to spatial patches of nodes on the sphere. (b) (b-i) Tuning curve shows that units in layer-2 have a preferred orientation. (c) Self-organizing networks on Poincare disks. (c-ii) Snapshot of a traveling bump. (c-iii) Receptive fields of units in layer-II.

**R3** raises a major concern regarding the novelty of this work as the emergence of complex feature-selective neurons has been described in the 90s. Our major contribution in this paper is towards demonstrating properties such as *flexibility*, *robustness* and *reconfigurability* of the developmental algorithm and showing that these properties allow us to "grow" artificial systems purely by local rules, which to the authors knowledge, hasn't been explored before. Flexibility is essential for growing networks on curved surfaces (fig-1a), useful for spherical image analysis. R3 recommends that we show **orientation selectivity**. Reconfigurability enables us to grow and self-organize units in layer-2 that are orientation selective, by altering the properties of the emergent traveling bump of activity (fig-1b). We plot a tuning curve to show that units have a **preferred orientation**. R3 mentions that the self-organized networks aren't "pooling" in the sense of CNN's. As the sensor-nodes in the input layer are not evenly spaced, the classical definition of pooling breaks down. We refer to **pooling**, when units across layer-2 are connected to similar sized spatial-patches of nodes in layer-I. Post self-organization a max/average operation can be applied by the unit in layer-2, making it max/average pooling. A tight regulation of spatial-patch size has been observed in our networks as shown in histogram fig-5e in the paper.

**R1**: (Roson and Bauer) show evidence of dimensionality reduction from retina to LGN. R1 also pointed out that the developmental algorithm could be used for **bio-inspired hardware**. We completely agree with this as we've begun adapting this algorithm for implementation on Loihi (Intel's neuromorphic hardware). Currently, to configure spiking neural nets (SNN) on Loihi to perform tasks, one needs to specify a hand-designed SNN topology, manually define neurons that belong to different layers and specify connections between layers (ref: Chit-Kwan & Wild). Instead of hand-programming these networks, our developmental algorithm would enable neuron clustering into different layers via the growing process and self-organize connections between these layers through spontaneous activity in the lower layers. This would provide a flexible and scalable way to program neuromorphic hardware. As suggested by R1, we shall make additions to the abstract and changes to the convolutional terminology.

**R2**: Spiking neurons have been used instead of neural masses to allow for future implementation of this algorithm on hardware dedicated for spiking networks (neuromorphic chips). R2 was concerned about the biological plausibility of global inhibition scheme. As the magnitude of inhibition decays exponentially with distance, a small network (2000 nodes) has every node inhibiting $\sim 90\%$ of the nodes in the network (as seen in insects). However, as we scale the network to 50000 nodes, every node inhibits $< 4\%$ of the nodes in the network, making it biologically feasible. We have also shown scaling of the developmental algorithm with the network size in supplementary (S5). R2 also suggested that we demonstrate the formation of pooling networks on a **Poincare disk**, with a non-uniform distribution of sensor-nodes. This is shown in Figure-1c. R2 wanted clarification on how random networks performed with high accuracy. Functionality of the network has been measured by connecting 2 layers of the network, either hand-crafted, grown from a single unit (our developmental algorithm), or random to a perceptron to perform the MNIST task. Only the perceptron was trained while keeping the first 2 layers fixed. This serves as a control to show that "grown" networks extract useful features and perform as well as hand-crafted networks (detailed in supplementary-S4). We have attempted to answer major comments and append results suggested as improvements, but are limited on space to answer all comments.

[Meta-Review · NeurIPS 2019]

The authors propose a developmental algorithm that grows a spiking neural architecture in a self-organized manner from local rules. A CNN-like retinotopic connectivity structure is emerging. The model is applied to several setups, including the MNIST dataset. They show that the grown network extracts useful features and can performs as well as hand-crafted networks. The work is very original and the results are interesting. The paper is well-written and technically sound. On the negative side, the terminology is not always clear and the results do not seem overly surprising. The authors submitted a strong Author's response which clarified many points and was well-received by the reviewers. The authors are urged to fix their vocabulary: describing the emergent receptive fields as "convolutional" (and, quite likely, "pooling") is inappropriate due to lack of weight-sharing; the correct term that should be used is "retinotopic".